# Endothelial Monocyte-Activating Polypeptide-II Is an Indicator of Severity and Mortality in COVID-19 Patients

**DOI:** 10.3390/vaccines10122177

**Published:** 2022-12-19

**Authors:** Manal Mohamed Saber, Azhar Mohamed Nomair, Ashraf M. Osman, Hanan Mohamed Nomeir, Naglaa M. Farag

**Affiliations:** 1Department of Clinical Pathology, Faculty of Medicine, Minia University, Minia 61519, Egypt; 2Department of Chemical Pathology, Medical Research Institute, Alexandria University, Alexandria 21561, Egypt; 3Medical Biochemistry Department, Faculty of Medicine, Alexandria University, Alexandria 21131, Egypt

**Keywords:** EMAP II, COVID-19, severity, mortality

## Abstract

Data for predicting the severity and mortality of coronavirus disease 2019 (COVID-19) are limited, and investigations are ongoing. Endothelial monocyte-activating protein II (EMAP-II) is a multifunctional polypeptide with pro-inflammatory properties. EMAP-II is a significant pathogenic component in chronic inflammatory lung diseases and lung injury. In this study, we aimed to assess the potential utility of EMAP-II as a predictor of COVID-19 severity and mortality. This study included 20 healthy volunteers and 60 verified COVID-19 patients. Nasopharyngeal samples from COVID-19-positive subjects and normal volunteers were collected at admission. The nasopharyngeal samples were subjected to EMAP-II real-time polymerase chain reaction (RT-PCR). EMAP-II RNA was not detected in nasopharyngeal swabs of normal controls and mild to asymptomatic COVID-19 patients and was only detectable in severe COVID-19 patients. EMAP-II critical threshold (Ct) was positively associated with lymphocyte percentages and oxygen saturation (*p* < 0.001) while being negatively associated with age (*p* = 0.041), serum CRP, ferritin, and D-dimer levels (*p* < 0.001). EMAP-II Ct cutoff ≤34 predicted a worse outcome in COVID-19 illness, with a sensitivity and specificity of 100%. Our study suggests that EMAP-II could be considered a potential biomarker of COVID-19 severity. EMAP-II can predict the fatal outcome in COVID-19 patients.

## 1. Introduction

Severe acute respiratory syndrome coronavirus 2 (SARS-CoV-2) is a novel virus that induces coronavirus illness 2019 (COVID-19) and has produced a global health crisis [1,2]. Although most COVID-19 patients have mild flu-like symptoms or may be asymptomatic, a small percentage of patients experience acute respiratory distress syndrome (ARDS), severe pneumonia, multi-organ failure, and sometimes death [3,4,5,6,7,8]. Although viremia and viral load are associated with the severity of COVID-19 infection [9,10,11,12], no accurate prognostic markers for predicting disease severity and mortality exist. Previous studies have reported that endothelial monocyte-activating protein II (EMAP-II) levels are increased in chronic inflammatory and obstructive lung disease [13,14] and that EMAP-II monoclonal antibody abrogated Influenza A Virus (IAV)-induced lung injury [15].

EMAP-II (P43, AIMP1, sync1) is a protein exhibiting pleiotropic effects on neutrophils, macrophages/monocytes, and endothelial cells, and which was initially identified in cultured methylcholanthrene A (meth A) murine fibrosarcoma cells [16,17,18]. EMAP-II triggers von Willebrand factor release, expression of P- and E-selectin in endothelial cells, activation of the neutrophil respiratory burst, and chemotaxis of macrophages and neutrophils [19]. EMAP-II activates normal dendritic cells and macrophages to enhance T-helper 1 responses and interleukin-12 release [20,21]. EMAP-II injection induces an inflammatory infiltration in the mouse footpad [19]. EMAP-II has also been shown to inhibit angiogenesis and enhance endothelial cell apoptosis [22,23]. These findings support the idea that EMAP-II is a proinflammatory peptide with anti-angiogenic effects.

EMAP-II is synthesized as a 34-kDa intracellular protein, which is cleaved to produce the extracellular mature form [19,24]. Behrensdorf et al. [25] reported that the 34-kDa form is vulnerable to caspase-7 cleavage, and some previous reports assume that additional proteases were involved in EMAP-II cleavage [26,27,28,29,30]. Quevillon et al. [31] showed that the hamster p43 molecule, a member of the multisynthase complex, is similar to human EMAP-II. They suggested 34-kDa EMAP-II may be a proteolytic product of p43 and that the human EMAP-II and p43 homologs coincide.

EMAP-II is found in the cellular cytoplasm [27,29,32,33,34,35], but stresses like viral infection, lipopolysaccharides, hypoxia, or apoptosis promote extracellular EMAP-II secretion and increase EMAP-II synthesis levels [36,37,38,39,40,41,42]. EMAP II might have immunosuppressive effects by inducing lymphocyte apoptosis [43,44,45,46]. Previous research has focused on the role of EMAP-II in virus-induced lung injury [15,41,47]. EMAP-II has become a novel target for treating lung disorders [15,48].

This study was carried out to investigate EMAP-II levels in COVID-19 subjects and the potential value of EMAP-II in predicting COVID-19 severity and mortality to guide COVID-19 clinical assessment.

## 2. Subjects and Methods

### 2.1. Study Population

The study comprised 60 COVID-19 patients from February 2022 to March 2022. Twenty healthy controls were also included. This prospective study was carried out at the two Minia University Hospitals in Egypt: Minia Cardiology and Chest University Hospital and Minia Liver University Hospital. The Minia Liver University Hospital was primarily designated for less severe patients, whereas the Minia Cardiology and Chest University Hospital is for severe cases.

### 2.2. Patients

A cohort of 60 volunteers confirmed to have been COVID-19 patients participated in this study. The patients’ datasets are attached in the Appendix A. The patients tested positive for COVID-19 via real-time quantitative polymerase chain reaction (RT-qPCR), following the guidelines of WHO for RT-qPCR [49]. Additionally, a physician validated the clinical symptoms and signs of COVID-19 infection. COVID-19 subjects were divided into three different groups: mild, asymptomatic, and severe [50], according to the National Health Commission of China’s guidelines for diagnosing COVID-19 illness. Patients with mild flu-like symptoms (loss of taste, smell, coughing, fever, or dyspnea) and with routine lung imaging were considered mild COVID-19 patients [51]. Severe COVID-19 patients were identified as subjects having the following criteria: respiratory rate ≥ 30 times/min with shortness of breath, PaO_2_/ oxygen concentration ≤300 mm Hg, and oxygen saturation ≤93% at rest. Patients were also classified as having a severe case of thoracic computerized tomography scans that showed >50% lesion progression [52,53].

The enrolled COVID-19 patients were monitored for 30 days or until they died to assess the clinical results. The evaluation excluded patients who asked to leave the trial, those with missing data, and those who passed away from causes other than ARDS.

The study excluded individuals with tumors, renal failure, liver diseases, immune-modulatory medication use, chronic respiratory illness, autoimmune diseases, cancer, kidney failure, liver dysfunction, lactation and pregnancy.

### 2.3. Laboratory Assay

All the subjects had peripheral blood collected under strict aseptic conditions. Blood samples were taken from all 60 COVID-19 patients and 20 healthy controls (8 mL each). For a complete blood count (CBC), 2 mL of blood was put into an EDTA tube and assessed using a CELLTAC Automatic Hematology Analyzer (G Nihon Kohden Corporation, Tokyo, Japan). A total of 1.6 mL of blood was put into a citrated tube (3.2% trisodium citrate) to measure the erythrocyte sedimentation rate (ESR), which was evaluated via the Westergren method. The last 4 mL of blood was put in a plain tube to clot, then centrifuged for serum isolation at room temperature. Then, the serum samples were separated and used. Serum levels of alanine aminotransferase (ALT), aspartate aminotransferase (AST), albumin, direct bilirubin, total bilirubin, creatinine, sugar, and blood urea were determined using a fully automated chemistry auto-analyzer system, Selectra Pro Xl (Eli Tech clinical system, Puteaux, France). A semiquantitative determination of C-reactive protein (CRP) was performed using the latex agglutination test. D-dimers and serum ferritin levels were measured using the Aia 360 automated immunoassay analyzer (Tosoh Biosciences, South San Francisco, CA, USA).

### 2.4. RNA Extraction and qRT-PCR

Individual nasopharyngeal swabs from COVID-19 subjects and normal controls were used to isolate EMAP-II RNA. Following WHO standards for qRT-PCR testing of nasopharyngeal samples was used to confirm COVID-19 infection [49]. Nasopharyngeal samples were taken using sterile polyester-tipped swabs. Each subject’s swabs (nasopharyngeal samples) were put into a single tube containing a universal medium. For sample manipulation, a biosafety class II laboratory was used.

EMAP-II RNA extraction was carried out using the RNAeay extraction kit (Qiagen, Hilden, Germany) in accordance with the manufacturer’s instructions. A fully automated QIAcube instrument (Qiagen, Hilden, Germany) was applied to extract EMAP-II RNA. Once the RNA was extracted, it was kept at 20 °C prior to the RT-PCR test. Reverse RNA transcription was performed using the QuantiTect reverse-transcription kit (Qiagen, Hilden, Germany). Reverse transcription was conducted under the subsequent circumstances: 42 °C for fifteen minutes and 95 °C for three minutes.

Target primers were acquired from (Qiagen, Hilden, Germany). The qRT-PCR was carried out utilizing SYBR green (Qiagen, Hilden, Germany). The following thermal cycling conditions were used: 2 min at 95 °C, 40 cycles of 5 s at 95 °C, and then 10 s at 60 °C. The melting curve was studied to determine every reaction’s melting point (mp) at 65–95 °C. Each test run contained negative controls (nuclease-free water). A sample was considered EMAP-II-positive if the EMP II gene’s amplification curve crossed the threshold line in less than 40 cycles. The EMAP-II expression was assessed in terms of the EMAP-II gene’s cycle threshold (Ct) value. The qRT-PCR test yielded a Ct value, identified as the number of amplification cycles necessary for a target gene to exceed the threshold line [54]. As the Ct value reflects the amount of genetic material (RNA) in the sample, a low Ct value was associated with a high gene expression [55].

### 2.5. Statistical Analysis

Data analysis was performed using the IBM SPSS 20.0 statistical program (IBM; Armonk, New York, NY, USA). The data were displayed as the median and interquartile range (IQR) for numerical data, in addition to both number and percentage for categorized data.

Differences between three or more groups concerning continuous variables were assessed utilizing the Kruskal–Wallis test. Mann–Whitney U-analysis for nonparametric data was conducted to compare the two independent groups. Chi-square or Fisher’s exact tests were performed to compare the categorical parameters. Pearson correlation was carried out for associations between the parameters. To determine the diagnostic utility of EMAP-II, receiver operating characteristic (ROC) curve analysis was performed. A *p*-value of less than 0.05 was regarded as significant.

## 3. Results

### 3.1. Demographic and Clinical Results

The current study enrolled a total of 80 subjects. Table 1 displays the subjects’ demographic and clinical traits. The severe COVID-19 subjects were significantly older, with a median age of 65.5 years, compared with the mild (median = 30.5 years) and asymptomatic groups (median = 35 years) (*p* < 0.001). The mild, asymptomatic, and severe patient groups comprised 11 males and 9 females each (*p* > 0.05). The healthy controls included 12 males and 8 females.

The analysis of COVID-19 symptoms and oxygen saturation among the three groups is shown in Table 1. The severe and mild groups showed significant cough, fever, and sore throat symptoms compared with controls (*p* < 0.001). Severe COVID-19 patients had marked dyspnea and low oxygen saturation compared to healthy controls (*p* < 0.001). We also found statistically relevant differences in cough, dyspnea, and lower oxygen saturation between patients with severe COVID-19 illness and the asymptomatic and mild subjects. Cough and sore throat were observed more frequently in the severe cases compared with asymptomatic cases (*p* < 0.001). Additionally, significant COVID-19 symptoms were observed in the mild COVID-19 patients when compared to asymptomatic patients (*p* < 0.05). However, there was no significant difference detected between asymptomatic and mild patient groups regarding dyspnea.

The number of symptoms varied significantly between the patient groups, with the severe COVID-19 subjects having more symptoms than the asymptomatic group or those with light symptoms (*p* < 0.001).

Furthermore, there were observed differences among the three groups regarding associated chronic illness. Patients with severe COVID-19 illness were more likely to have hypertension and diabetes mellitus compared with mild and asymptomatic patients (*p* < 0.05). The percentage of diabetic patients in the severe COVID-19 patient group increased significantly to 50%, compared to 20% in the mild COVID-19 patient group and 25% in the asymptomatic COVID-19 patient group. The prevalence of hypertension ranged from 50% in the severe COVID-19 subjects to 30% in the mild group and 10% in the asymptomatic group. The most prevalent comorbidities were asthma, benign prostatic hypertrophy, coronary heart diseases, cholecystitis, deep venous thrombosis, renal diseases, stroke, and thyroid diseases (Table 1).

In terms of clinical outcomes in the severe COVID-19 patient group, 10 recovered from ARDS and survived, while the remaining 10 died of ARDS. The remaining asymptomatic and mild COVID-19 cases survived (*p* < 0.001), Table 1.

### 3.2. Laboratory Characteristics

Table 2 illustrates the laboratory criteria in all studied groups. When comparing the mild, asymptomatic, and severe COVID-19 subjects to the healthy volunteers, a significant difference between COVID-19 patients and controls was observed with regard to the laboratory parameters (*p* < 0.05). However, no significant difference was observed regarding direct bilirubin, total bilirubin levels, and platelet count (*p* > 0.05).

Blood urea, serum creatinine, AST, direct bilirubin, random blood sugar, first and second ESR, D-dimers, and CRP were significantly higher in severe COVID-19 subjects than in normal volunteers (*p* < 0.05). In addition, the ferritin levels of severe COVID-19 subjects were significantly higher than those of healthy volunteers (*p* < 0.001). Moreover, compared to mild and asymptomatic patients, blood urea, serum creatinine, AST, first and second ESR, CRP, ferritin, and D-dimer levels were significantly increased in the severe COVID-19 subjects (*p* < 0.001). Furthermore, compared to asymptomatic groups, ALT levels were significantly increased in patients with severe COVID-19 illness (*p* = 0.004). Severe COVID-19 patients’ random blood sugar levels were significantly higher than those of asymptomatic individuals (*p* = 0.006).

Hemoglobin and serum albumin levels were significantly decreased in the severe, mild, and asymptomatic COVID-19 groups compared to normal volunteers (*p* < 0.001). Furthermore, when compared to controls, lymphocyte percentages were significantly lower in mild, asymptomatic, and severe COVID-19 subjects (*p* < 0.001). Compared to the asymptomatic and mild groups, the severe COVID-19 patient group had significantly lower lymphocyte percentages and albumin levels (*p* < 0.001). Furthermore, severe COVID-19 patients’ hemoglobin levels were significantly lower than those of mild and asymptomatic individuals (*p* = 0.017 and *p* < 0.001, respectively).

### 3.3. Detectable EMAP-II RNA in Severe COVID-19 Patients

We assessed EMAP-II RNA in the nasopharyngeal swabs from all subjects. Table 2 shows the EMAP-II expression in all studied groups. We did not find EMAP-II expression in the controls or the COVID-19 patients with mild and asymptomatic symptoms. We could only detect EMAP-II in 20 patients with severe symptoms out of 80 samples (25%). The median EMAP-II Ct values were 34.3 (IQR: 32.9–36.1) in the severe COVID-19 group.

The severely ill COVID-19 patients were divided into high- and low-EMAP-II Ct groups (≤34 vs. >34) (Table 3). Among the 20 positive patients, we found EMAP-II Ct ≤ 34 in 10 patients (50%) and EMAP-II Ct > 34 in the other 10 patients (50%). The median Ct value of EMAP-II in the EMAP-II Ct ≤ 34 group was 32.9 (IQR: 31.6–34), while the median EMAP-II Ct value in the EMAP-II Ct > 34 group was 36.1 (IQR: 35.1–36.5) (*p* < 0.001). Ten patients died in the EMAP-II Ct ≤ 34 group, compared with ten patients who survived in the EMAP-II Ct > 34 group (*p* < 0.001).

Age tended to be higher in the EMAP-II Ct ≤ 34 group compared with the EMAP-II Ct > 34 group (*p* = 0.007). The EMAP-II Ct ≤ 34 groups had one patient younger than 60 and nine patients over 65 years old. The EMAP-II Ct > 34 group comprised seven patients younger than 60 and three patients over 65 years old (*p* = 0.020) (Table 3).

D-dimer, CRP, and ferritin levels were significantly higher in the EMAP-II Ct ≤ 34 group compared with the EMAP-II Ct > 34 group (*p* < 0.001, *p* = 0.011 and *p* < 0.001) (Table 3), while lymphocyte percentage and O_2_ saturation were significantly lower in the EMAP-II Ct ≤ 34 group compared to the EMAP-II Ct > 34 group (*p* < 0.001).

### 3.4. Association between EMAP-II and Clinical and Laboratory Parameters

We studied severe COVID-19 subjects’ clinical and laboratory overviews to see if there was any potential association between EMAP-II and disease progression. Table 4 illustrates the relationships between all parameters and EMAP-II in the severe COVID-19 patient group. EMAP-II Ct revealed significant positive associations with lymphocyte percentage and O2 saturation in the severe COVID-19 patient group (*p* < 0.001). EMAP-II Ct showed a negative relationship with ferritin, CRP, and D-dimer levels (*p* < 0.001). Moreover, EMAP-II Ct showed a negative association with age (*p* = 0.023).

### 3.5. Diagnostic Utility of EMAP-II for Mortality in COVID-19 Patients

The ROC analysis produced significant results in the severe COVID-19 patient group, as illustrated in Figure 1. The ROC study demonstrated that EMAP-II is predictive of fatal outcomes. EMAP-II had high sensitivity and specificity of 100% with AUC 1.0 for risk prediction. The cutoff value of EMAP-II Ct was ≤34, which significantly differentiated severe COVID-19 infections. Collectively, EMAP-II allowed the identification of COVID-19 patients at high risk for fatal outcomes.

## 4. Discussion

In this study, we assessed EMAP-II RNA in nasopharyngeal swabs in COVID-19 patients and revealed that EMAP-II is involved in COVID-19 severity and mortality. Our study demonstrated that the normal healthy controls did not express EMAP-II RNA within their nasopharyngeal samples. EMAP-II RNA transcripts are not detected in the normal respiratory tract [29]. Detection of EMAP-II RNA expression is restricted to the brain, testis, and thymus [38]. Subsequently, early viral infections are not accompanied by an increase in EMAP-II RNA levels or EMAP-II intracellular expression [15]. Minor and asymptomatic illnesses have effective innate immune responses, giving the T-cell response time to develop. Therefore, EMAP-II expression is controlled [56].

Our results suggest that EMAP-II is only present in severe COVID-19 patients, but the exact mechanism by which this happens is not fully understood. EMAP-II could serve as a biomarker for viral infections [42]. High EMAP-II expression in severe COVID-19 patients might reflect the inflammatory role of this protein. EMAP-II has been linked to viral lung damage via epithelial and endothelial apoptosis induction [13,15,41,47]. EMAP-II might have a significant role in exacerbating COVID-19 infection by inducing lymphocyte apoptosis. Diseased lung tissue might release EMAP-II, resulting in EMAP-II-induced lymphocyte death. Additional study is needed to support this idea.

A possible explanation for EMAP-II expression in severe COVID-19 patients is increased intracellular EMAP-II synthesis. Hypoxia and apoptosis enhance EMAP-II expression both in vivo and in vitro [36,38]. Hypoxia is a sign of lung involvement in SARS-CoV-2 [57,58]. COVID-19 patients have displayed elevated carbonic anhydrase activity as a marker for hypoxia [59,60]. Youssef et al., reported an association between EMAP-II and carbonic anhydrase [44]. Hypoxia is known to upregulate the release and expression of matrix metalloproteinases and plasminogen activator-1. These enzymes may be involved in enhancing EMAP-II expression [43]. Programmed cell death (PCD) pathways are frequently found upstream of inflammatory processes and may be crucial in mediating severe COVID-19 disease [61], which might result in EMAP-II expression. Further experiments are needed to explore these theories.

Our results revealed that COVID-19 patients with lymphopenia, lower oxygen saturation, and high CRP, ferritin, and D-dimer levels had lower EMAP-II Ct values (≤34) than patients without these symptoms (EMAP-II Ct values > 34). A low Ct value is related to increased gene expression since it measures the amount of genetic material (RNA) in the sample [55]. Severe COVID-19 patients might display high EMAP-II expression, which results in direct damage and apoptosis of respiratory lining cells and the alveolar epithelium. The results revealed a considerably higher EMAP-II expression in COVID-19 subjects with poor prognoses than in cases with favorable development.

Our results indicate that EMAP-II expression and inflammatory mediators, such as CRP, ferritin, and D-dimers, were significantly correlated in severe COVID-19 patients. Our findings were in line with a previous publication which showed that high EMAP-II expression was associated with inflammation in coronavirus infection [62]. EMAP-II draws neutrophils to infected areas [16,63], participating in the immune response during viral infections’ inflammatory processes [40]. COVID-19 pathogenesis heavily relies on the inflammatory response [64], while ferritin, CRP, and D dimers are associated with mortality in COVID-19 patients [65]. EMAP-II might represent later pathological changes in the lower airway associated with ARDS. Significant lymphopenia was found to be linked to EMAP-II. The COVID-19 virus induces lymphopenia by enhancing lymphocyte apoptosis in severe COVID-19 patients [66,67]. Previous research has indicated that EMAP-II triggers lymphocyte apoptosis, pointing to an immunosuppressive function. EMAP-II inhibitors may affect the apoptotic cascade and understanding this could usher in a new era of COVID-19 therapeutics.

Our study is the first to investigate EMAP-II’s role in COVID-19 mortality prediction. ROC curve analysis showed high accuracy of EMAP-II (AUC = 1.0) for predicting COVID-19 mortality. COVID-19 patients who died had a lower EMAP-II Ct (≤34) than those who survived. Our results found higher EMAP-II expression in COVID-19 patients with poor prognoses than in cases with a favorable development. EMAP-II might reflect the critical deterioration of COVID-19 illness. The data suggest that EMAP-II might be a valuable biomarker of COVID-19 disease outcomes.

## 5. Study Limitations

The current study has some limitations. Our sample size was too small to assess the best variable cutoffs and establish suitable predictive values of EMAP-II, so additional large studies are required to validate our findings. Furthermore, to assess EMAP-II’s role as a pathological marker of COVID-19, further studies involving biopsy, autopsy, or bronchoalveolar analysis should be carried out. A prospective, large-cohort study is required to identify the clinical significance of EMAP-II as a patient stratification tool.

## 6. Conclusions

Our results identified EMAP-II as a promising marker for predicting COVID-19 illness severity, and mortality outcomes. Furthermore, high EMAP II expression in severe COVID-19 patients was associated with poor outcomes. The study offers significant data for clinicians to quickly identify COVID-19 individuals at risk for illness progression and fatal outcomes. This supports the use of EMAP-II as a prognostic marker for the detection of severe COVID-19 subjects with a high risk for unfavorable outcomes. Due to COVID-19’s evidence of inflammation, the assessment of a proinflammatory polypeptide, EMAP-II, provides accurate predictive information on the COVID-19 illness. Further studies assessing multi-markers with EMAP-II could offer much information for predicting COVID-19 severity and mortality risk. Future studies, including an extended clinical assessment, might help to better understand EMAP-II’s role in COVID-19 disease.

## Figures and Tables

**Figure 1 vaccines-10-02177-f001:**
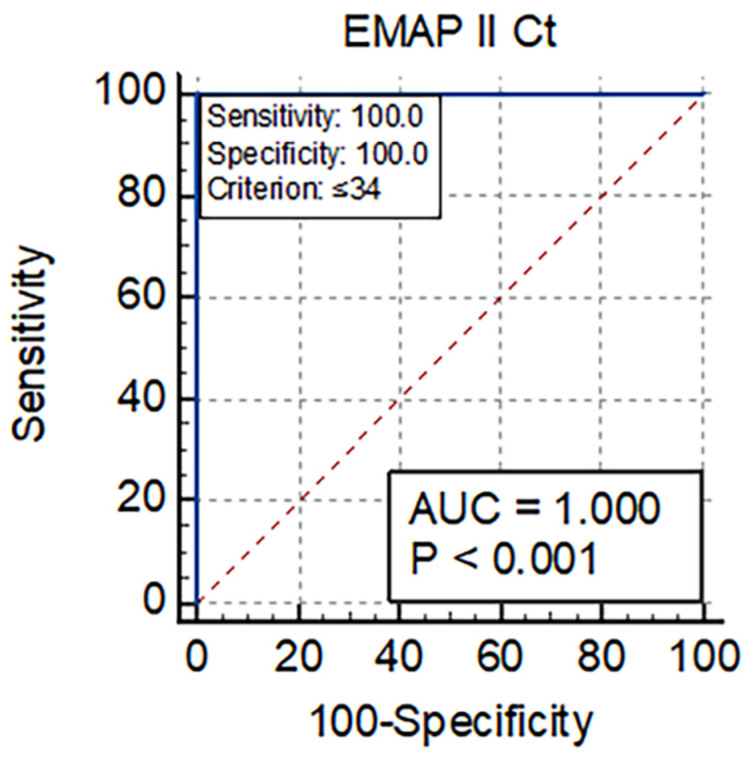
ROC curves of EMAP II as predictors of mortality in the severe COVID-19 patient’s group.

**Table 1 vaccines-10-02177-t001:** Comparison of demographic and clinical criteria in the studied groups.

	Severe (I)	Mild (II)	Asymptomatic (III)	Control	*p*-Value ^a^			
	(N = 20)	(N = 20)	(N = 20)	(N = 20)		*p*1 (I vs. II)	*p*2 (I vs. III)	*p*3 (II vs. III)
Age	65.5 (56.5–68)	30.5 ^‡^ (27.5–53.5)	35 ^‡^ (26.5–48)	63 (57.5–66)	<0.001 **	<0.001 **	<0.001 *	0.542
<65 year	8 (40%)	17 (85%)	20 (100%)	12 (60%)	<0.001 **	0.003 *	<0.001 **	0.231
≥65 year	12 (60%)	3 (15%)	0 (0%)^‡^	8 (40%)				
Sex								
Male	11 (55%)	11 (55%)	11 (55%)	12 (60%)	0.985	<0.99	<0.99	<0.99
Female	9 (45%)	9 (45%)	9 (45%)	8 (40%)				
Chronic disease								
Asthma	1 (5%)	0 (0%)	0 (0%)	0 (0%)	0.388	0.567	0.439	0.615
BPH	1 (5%)	0 (0%)	0 (0%)	0 (0%)				
Cardiac disease	3 (15%)	2 (10%)	1 (5%)	0 (0%)				
Cholecystitis	0 (0%)	1 (5%)	0 (0%)	0 (0%)				
DVT	1 (5%)	0 (0%)	0 (0%)	0 (0%)				
Renal disease	1 (5%)	1 (5%)	2 (10%)	0 (0%)				
Stroke	1 (5%)	0 (0%)	0 (0%)	0 (0%)				
Thyroid disease	0 (0%)	0 (0%)	1 (5%)	0 (0%)				
Hypertension	10 (50%) ^‡^	6 (30%) ^‡^	2 (10%)	0 (0%)	<0.001 **	0.197	0.006 *	0.235
Diabetes Mellitus	10 (50%) ^‡^	4 (20%)	5 (25%)	0 (0%)	0.003 *	0.047 *	0.102	>0.99
Cough	19 (95%) ^‡^	16 (80%) ^‡^	0 (0%)	0 (0%)	<0.001 **	0.342	<0.001 **	<0.001 **
Fever	18 (90%) ^‡^	12 (60%) ^‡^	0 (0%)	0 (0%)	<0.001 **	0.028 *	<0.001 **	<0.001 **
Dyspnea	18 (90%) ^‡^	2 (10%)	0 (0%)	0 (0%)	<0.001 **	<0.001 **	<0.001 **	0.487
Loss of smell and taste	4 (20%)	7 (35%) ^‡^	0 (0%)	0 (0%)	0.002 *	0.288	0.106	0.008 *
Sore throat	14 (70%) ^‡^	10 (50%) ^‡^	0 (0%)	0 (0%)	<0.001 **	0.197	<0.001 **	<0.001 **
Number of symptoms	4 ^‡^ (3–4)	3 ^‡^ (2–3)	0 (0–0)	0 (0–0)	<0.001 **	<0.001 **	<0.001 **	<0.001 **
O2 Saturation	73.5 ^‡^ (61–84.5)	96 (95–97.5)	98 ^‡^ (97–98)	97 (96–97.5)	<0.001 **	<0.001 **	<0.001 **	0.009 *
Outcome								
Survived	10 (50%) ^‡^	20 (100%)	20 (100%)	20 (100%)	<0.001 **	<0.001 **	<0.001 **	…
Died	10 (50%)	0 (0%)	0 (0%)	0 (0%)				

BPH—benign prostate hypertrophy; DVT—deep venous thrombosis. * Statistics were considered significant at *p* < 0.05. ** Statistics were considered highly significant at *p* < 0.001. ^a^
*p*-value between the four groups. ^‡^ Statistically significant difference compared to the control group. Quantitative data are expressed as median (IQR—interquartile range).

**Table 2 vaccines-10-02177-t002:** Comparison of laboratory criteria and EMAP II Ct in the studied groups.

	Severe (I)	Mild (II)	Asymptomatic (III)	Control	*p*-Value ^a^			
	(N = 20)	(N = 20)	(N = 20)	(N = 20)		*p*1 (I vs. II)	*p*2 (I vs. III)	*p*3 (II vs. III)
Hb (g/dl)	12.2 ^‡^ (11.2–13.2)	13.1 ^‡^ (12.6–14.2)	14.1 (13.1–15.5)	14.1 (13.7–15.1)	<0.001 **	0.017 *	<0.001 **	0.076
TLC × 10^3^/mm^3^	12.1 ^‡^ (8.6–13.1)	6 ^‡^ (4–7)	5.7 ^‡^ (4.1–7.5)	7.6 (6.3–8.9)	<0.001 **	<0.001 **	<0.001 **	0.818
PLT × 10^3^/ mm^3^	277 (223–338)	273 (214–333)	342 (242–381)	309 (268–349)	0.176	0.715	0.172	0.066
Lymphocyte percentage	10 ^‡^ (6.5–12)	17 ^‡^ (14–21.5)	14.5 ^‡^ (13–17)	35.5 (30–41)	<0.001 **	<0.001 **	<0.001 **	0.103
ESR 1st	55.5 ^‡^ (40.5–90)	30 ^‡^ (23–41)	23 ^‡^ (18.5–28)	6 (5–7)	<0.001 **	<0.001 **	<0.001 **	0.008 *
ESR 2nd	99.5 (83.5–130) ^‡^	64.5 ^‡^ (50–76.5)	45 ^‡^ (39.5–55)	13 (10.5–15)	<0.001 **	<0.001 **	<0.001 **	0.006 *
Urea (mg/dL)	55.5 ^‡^ (38.5–69.5)	33.5 (30–38)	27 (24–31.5)	30 (28.5–33.5)	<0.001 **	0.001 *	<0.001 **	0.003 *
Creatinine (mg/dL)	1.2 ^‡^ (1.1–1.4)	0.9 (0.7–1.1)	0.8 (0.7–0.9)	0.9 (0.8–0.9)	<0.001 **	0.002 *	<0.001 **	0.068
ALT (U/L)	38 ^‡^ (27.5–62)	31 ^‡^ (24–37.5)	24.5 (20–33.5)	22 (18.5–28)	<0.001 **	0.096	0.004 *	0.07
AST (U/L)	38.5 (29.5–46.5) ^‡^	26 (22.5–29.5)	21.5 (17.5–31)	24 (19–29.5)	<0.001 **	<0.001 **	<0.001 **	0.249
Albumin (g/dL)	3.7 ^‡^ (3.5–3.9)	4.1 (4–4.3)	4.2 (4–4.4)	4.1 (3.9–4.4)	<0.001 **	<0.001 **	<0.001 **	0.173
Total bilirubin (mg/dL)	0.8 ^‡^ (0.6–0.9)	0.6 (0.5–0.9)	0.7 (0.5–0.8)	0.6 (0.5–0.8)	0.133	0.166	0.068	0.722
Direct bilirubin (mg/dL)	0.2 ^‡^ (0.2–0.3)	0.2 (0.1–0.2)	0.2 (0.1–0.2)	0.2 (0.1–0.2)	0.078	0.235	0.102	0.654
RBS (mg/dl)	145.5 ^‡^ (119.5–195)	121.5 (104–139)	115 (94.5–131.5)	115.5 (103–126)	0.009 *	0.107	0.006 *	0.273
D-dimer (µg/mL)	6 ^‡^ (3.3–10.5)	1.3 ^‡^ (0.9–1.8)	0.5 ^‡^ (0.4–0.6)	0.3 (0.2–0.3)	<0.001 **	<0.001 *	<0.001 **	<0.001 **
Ferritin (ng/mL)	923 ^‡^ (642.5–1362.5)	431.5 ^‡^ (370–540)	207 ^‡^ (138–328)	106.5 (79–124.5)	<0.001 **	<0.001 **	<0.001 **	<0.001 **
CRP (mg/L)	130 (105–198) ^‡^	23 ^‡^ (15.9–33.9)	16.3 ^‡^ (12.7–18.6)	2 (1.6–2.2)	<0.001 **	<0.001 **	<0.001 **	0.012 *
EMAP II Ct	34.3 (32.9–36.1)							

ESR—erythrocyte sedimentation rate; TLC—total leukocytic count; Hb—hemoglobin; PLT—platelet count; RBS—random blood sugar; Ct—critical threshold; ALT—alanine aminotransferase; AST—aspartate aminotransferase. * Statistics were considered significant at *p* < 0.05. ** Statistics were considered highly significant at *p* < 0.001. ^a^
*p*-value between the four groups. ^‡^ Statistically significant difference compared to the control group. Quantitative data are expressed as median (IQR—interquartile range).

**Table 3 vaccines-10-02177-t003:** Comparison of low and high EMAP II Ct in the severe COVID-19 patient’s group.

	EMAPII Ct ≤ 34	EMAPII Ct > 34	*p* Value
	(N = 10)	(N = 10)	
Mortality	10 (100%)	0 (0%)	<0.001 **
Age	67.5 (66–70)	58 (55–65)	0.007 *
<60 year	1 (10%)	7 (70%)	0.020 *
≥65 year	9 (90%)	3 (3%)	
Sex			
Male	4 (40%)	7 (70%)	0.37
Female	6 (60%)	3 (30%)	
Hypertension	5 (50%)	5 (50%)	>0.99
Diabetes Mellitus	6 (60%)	4 (40%)	0.371
Cough	10 (100%)	9 (90%)	>0.99
Fever	9 (90%)	9 (90%)	>0.99
Dyspnea	10 (100%)	8 (80%)	0.474
Loss of smell and taste	3 (30%)	1 (10%)	0.582
Sore throat	6 (60%)	8 (80%)	0.628
Number of symptoms	4 (3–5)	4 (3–4)	0.304
O2 saturation	61 (60–70)	84.5 (80–86)	<0.001 **
Hb (g/dl)	12 (11.4–12.7)	12.5 (10.8–13.5)	0.733
TLC × 10^3^/mm^3^	11.2 (9.9–12.5)	12.6 (8.1–14.2)	0.52
PLT × 10^3^/mm^3^	273 (176–362)	304 (246–334)	0.705
Lymphocyte percentage	6.5 (5–8)	12 (11–13)	<0.001 **
ESR 1st	60.5 (45–90)	50.5 (40–90)	0.52
ESR 2nd	107 (94–135)	92.5 (78–125)	0.52
Urea (mg/dl)	63.5 (49–70)	50.5 (38–57)	0.345
Creatinine (mg/dl)	1.3 (1.1–1.4)	1.1 (1–1.2)	0.147
ALT (U/L)	40 (27–66)	38 (29–55)	0.97
AST (U/L)	37 (29–48)	38.5 (34–43)	0.733
Albumin (g/dl)	3.6 (3.6–3.7)	3.9 (3.3–4)	0.249
Total bilirubin (mg/dl)	0.8 (0.6–0.8)	0.8 (0.6–1)	0.415
Direct bilirubin (mg/dl)	0.2 (0.2–0.2)	0.2 (0.2–0.3)	0.297
RBS (mg/dl)	148.5 (110–180)	139 (128–220)	0.705
D-dimer (ug/mL)	10.5 (6.4–10.9)	3.3 (3.2–4.9)	<0.001 **
Ferritin (ng/mL)	1346 (941–1434)	642.5 (591–812)	<0.001 **
CRP (mg/L)	198 (175–380)	115.5 (102–128)	0.011 *
EMAP II Ct	32.9 (31.6–34)	36.1 (35.1–36.5)	<0.001 **

ESR—erythrocyte sedimentation rate; TLC—total leukocytic count; Hb—hemoglobin; PLT—platelet count; RBS—random blood sugar; Ct—critical threshold; ALT—alanine aminotransferase; AST—aspartate aminotransferase; * Statistics were considered significant at *p* < 0.05. ** Statistics were considered highly significant at *p* < 0.001. Quantitative data are expressed as median (IQR—interquartile range).

**Table 4 vaccines-10-02177-t004:** Correlation between EMAP II Ct with all parameters in the severe COVID-19 patient’s group.

	EMAP II Ct	
	rho	*p*
Age	−0.505	0.023 *
Number of symptoms	−0.162	0.494
O2 saturation	0.968	<0.001 **
Hb (g/dL)	0.079	0.741
TLC × 10^3^/mm^3^	0.123	0.605
PLT × 10^3^/mm^3^	0.212	0.37
Lymphocyte percentage	0.836	<0.001 **
ESR 1st	0.111	0.643
ESR 2nd	0.082	0.73
Urea (mg/dL)	0.065	0.786
Creatinine (mg/dL)	−0.105	0.659
ALT (U/L)	−0.13	0.585
AST (U/L)	0.023	0.922
Albumin (g/dL)	0.026	0.915
T bilirubin (mg/dL)	0.175	0.46
D bilirubin (mg/dL)	0.279	0.234
RBS (mg/dL)	0.158	0.505
D dimer (µg/mL)	−0.898	<0.001 **
Ferritin (ng/mL)	−0.833	<0.001 **
CRP (mg/L)	−0.725	<0.001 **

ESR—erythrocyte sedimentation rate; TLC—total leukocytic count; Hb—hemoglobin; PLT—platelet count; RBS—random blood sugar; Ct—critical threshold; ALT—alanine aminotransferase; AST—aspartate aminotransferase; * Statistics were considered significant at *p* < 0.05. ** Statistics were considered highly significant at *p* < 0.001.

## Data Availability

The datasets used in this research analysis are attached in the Appendix A of the manuscript.

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
