# Peer review of "Endothelial Monocyte-Activating Polypeptide-II Is an Indicator of Severity and Mortality in COVID-19 Patients"

_vaccines, 2022, doi:10.3390/vaccines10122177_

Round 1

Reviewer 1 Report

Dear Editor I reviewed the manuscript mentioned blow.

 EMAP II is an indicator of severity and mortality in patients 2 with SARS‑CoV‑2 infection

The authors investigated in 60 verified COVID-19 patients the prognostic value of EMAP II for the disease severity. For comparison, the authors undertake the testing in 20 healthy volunteers. In volunteers and in mild to asymptomatic COVID-19 patients EMAP II was not detected, while it was the case exclusively in 20 severe affected COVID-19 subjects. The authors calculated further associations in patients with a EMAP II positive finding.       

In summary, I recommend to publish the manuscript in Vaccines. It contains valuable findings with implications for the clinical practice.  

However, I have some major points of criticisms, which might help to some improvement:

1. I would give some more background information about EMAP II in the abstract. The explanation EMAP II is an endothelial monocyte-activating protein II (EMAP II) as a predictor of COVID-19 severity and mortality appears to me to less.  

2. In the abstract the authors took CT or Ct as abbreviation for cycle threshold. I would recommend keeping one version consistently throughout the manuscript.

3. In the abstract, do not use interpreting attributes in the results presentation such as interesting. The sentence Interestingly, a cutoff value of EMAP II Ct < 34 could differentiate…. should be reformulated. For example, a cut of <34 predicted in 100% of the cases a worse outcome; all patients with these values died due to COVID-19.      

4. line 51 in the manuscript. IAV is not defined previously in the text. Similar, there a lot of abbreviations, which need to be explained before the abbreviation is used.  

5. The analyses a comprehensive, the results are presented and calculated well.

6. I do not understand why so many authors start the discussion with a new comprehensive introduction. This is propaedeutic, therefore at this point this is only trash. Please delete line 300-318.

7. I would suggest to reformulate the entire discussion and focus on the presented results and put them into the context of the current literature. I am not always sure when reading it, what is meant. For example: Line 342: in early research we have …; what does that mean , some other data which are already published, what is the connection to the present manuscript. I would use terms such as: our results indicate, in our study we detected….  

8. Probably the manuscript will benefit from a review of a native English speaker.

Author Response

Special editing of the English language and style was performed by a special MDPI editor

All sections of MS are improved

1-I would give some more background information about EMAP II in the abstract. The explanation EMAP II is an endothelial monocyte-activating protein II (EMAP II) as a predictor of COVID-19 severity and mortality appears to me to be less.  

Comment 1: Many thanks for the valuable comments of the reviewer. Background information about EMAP II was added to the abstract.

2- In the abstract, the authors took CT or Ct as an abbreviation for cycle threshold. I would recommend keeping one version consistent throughout the manuscript.

Comment 2: Thanks for this comment; Ct as an abbreviation for cycle threshold was consistent throughout the manuscript.

  1. In the abstract, do not use interpreting attributes in the results presentation such as interesting. The sentence Interestingly, a cutoff value of EMAP II Ct < 34 could differentiate…. should be reformulated. For example, a cut of <34 predicted in 100% of the cases a worse outcome; all patients with these values died due to COVID-19. 

Comment 3: 3. In the abstract, interpreting attributes are deleted. a cutoff value of EMAP II Ct < 34 could differentiate was reformulated: EMAP-II Ct cutoff ≤ 34 predicted a worse outcome in COVID-19 illness, with a sensitivity and specificity of 100%.

  1. line 51 in the manuscript. IAV is not defined previously in the text. Similarly, there are a lot of abbreviations, which need to be explained before the abbreviation is used.  

Comment 4: IAV is defined in the text. Similarly, the abbreviations are explained and revised in the manuscript. A list of abbreviations is included at the end of the manuscript.

5- I do not understand why so many authors start the discussion with a new comprehensive introduction. This is propaedeutic, therefore at this point, this is only trash. Please delete lines 300-318.

Comment 5: lines 300-318 are deleted and the discussion was modified.

6- The analyses a comprehensive, the results are presented and calculated well.

Comment 6: Many thanks for this comment

7- I would suggest reformulating the entire discussion and focusing on the presented results and putting them into the context of the current literature. I am not always sure when reading it, what is meant. For example: In line 342: in early research we have …; what does that mean, some other data which are already published, what is the connection to the present manuscript. I would use terms such as our results indicate, in our study we detected….  

Comment 7: the entire discussion is formulated and focused on the presented results. In line 342, this sentence meant an earlier study Youssef, M. M. S.; Symonds, P.; Ellis, I. O.; Murray, J. C. EMAP-II-dependent lymphocyte killing is associated with hypoxia in colorectal cancer. British Journal of Cancer 2006, 95(6), 735–743. This was an earlier study by Manal Mohamed Saber about the association between EMAP II and hypoxia. Terms such as our results indicate, in our study we detected, were used  

8- Probably the manuscript will benefit from a review by a native English speaker.

Comment 8: the manuscript was reviewed by MDPI’s special English editor.  The certificate was sent to the editor.

Reviewer 2 Report

Comments:

1. Could the authors provide the ethical approval for their study please?

2. Could the authors please clarify how the numbers of study participants were determined i.e. how was their study powered?

3. Could the authors please provide the evidence for the normality of their data by their test please? As far as I am aware the test that they used may not be the best test for such small numbers of study participants.

Author Response

Special editing of the English language and style was performed by a special MDPI editor

All sections of MS are improved

  1. Could the authors provide ethical approval for their study, please?

Comment 1:  Thanks for this valuable comment the authors provide the whole ethical approval for the study, sent to the editor

  1. Could the authors please clarify how the number of study participants was determined i.e. how was their study powered?

Comment 2: In previous EMAP-II studies and publications, we know that EMAP-II RNA will not be found in the normal respiratory system. Also, EMAP-II RNA could not be detected in early viral infections. It could be detected only later when lung damage and acute lung injury happen. Due to the large-effect size between the different groups because of the presence of EMAP-II RNA in severe viral inflammatory lung illness (100%) and the absence of EMAP-II in controls and mild viral infections (0%), small size can be applied to this study

  1. Could the authors please provide evidence for the normality of their data by their test? As far as I am aware the test that they used may not be the best test for such small numbers of study participants.

Comment 3: Here, we appreciate the reviewer’s opinion of the test used. Data were presented as the median and interquartile range (IQR) for numerical data, in addition to both number and percentage for categorized data. The tests used are included in the statistical analysis section.

Statistical Analysis

The IBM SPSS 20.0 statistical package was used to analyze the data (IBM; Armonk, New York, NY, USA). Data were presented as the median and interquartile range (IQR) for numerical data, in addition to both number and percentage for categorized data. Differences between three or more groups concerning continuous variables were assessed utilizing the Kruskal-Wallis test. The Mann–Whitney U analysis for nonparametric data was conducted to compare the two independent groups. Fisher’s exact or chi-square tests were used to compare the categorical parameters. Pearson correlation was carried out for associations between the parameters. Receiver operating characteristic (ROC) curve analysis was carried out to identify EMAP-II diagnostic utility. A p-value of less than 0.05 was considered significant.

Reviewer 3 Report

This manuscript is on EMAP II and its potential to serve as an indicator on COVID-19 patient´s outcomes. Although the general topic is relevant to the COVID crisis, there are many shortcoming of this article that have to be adressed to provide significant context to readers.

Only SOME aspects are listed here:

Please note that authors should prepare the manuscript according to authors guidelines. E.g. The abstract should be a total of about 200 words maximum. The abstract should be a single paragraph and should follow the style of structured abstracts, but without headings. This is not done in the current manuscript.

The title should avoid abbreviations, if possible.

The introduction should be more informative on the rationale behind the study.

There are several phrases with an unclear meaning such as “The mild, asymptomatic, and severe patient groups comprised 11 males and nine females (p > 0.05), while the controls had similar numbers of females and males.”

Redundant information in the different chapters should be reduced.

Table 5 is too small to be a table. The information could be presented in the text.

Consistent use of terms such as COVID vs. Covid vs. covid is highly recommended.

Unfortunately, there several formatting problems regarding color and typos, also unfinished phrases such as in lines 105 or 356.

The authors state that before hospital admission, a reliable and straightforward biomarker would have a direct and immediate impact. But this impact is not described further and practical and theoretical implications are almost missing.

It seems that a full ethics proposal is missing for this study, altough it is a clinical study.

Why are the datasets used in this research's analysis not accessible to the general public? This brings a barrier to reproducibility of the findings by independent research organizations.

Are the findings transferable to non-Egyptian patients?

The conclusions are very weak. Also, mortality of COVID-19 patients might be mortality risk of COVID-19 patients?

Author Response

Reviewer 3

Special editing of the English language and style was performed by a special MDPI editor

All sections of MS are improved

1- Please note that authors should prepare the manuscript according to the authors’ guidelines. E.g. The abstract should be a total of about 200 words maximum. The abstract should be a single paragraph and should follow the style of structured abstracts, but without headings. This is not done in the current manuscript.

Comment 1:  Thanks for this valuable comment, the abstract is a total of about 200 words. The abstract is a single paragraph without headings.

2- The title should avoid abbreviations, if possible

Comment 2:  The title is now without the EMAP II abbreviation.

3- There are several phrases with an unclear meaning such as “The mild, asymptomatic, and severe patient groups comprised 11 males and nine females (p > 0.05), while the controls had similar numbers of females and males.”

Comment 3:  The phrases with unclear meanings are all corrected

4- Redundant information in the different chapters should be reduced.

Comment 4:  all redundant information in the different chapters is reduced.

5-Table 5 is too small to be a table. The information could be presented in the text.

Comment 5:  Table 5 is deleted. The information is presented in the text with the figure.

6- Consistent use of terms such as COVID vs. Covid vs. covid is highly recommended.

Comment 6:  the term COVID is consistent in the manuscript.

7- Unfortunately, there are several formatting problems regarding color and typos, also unfinished phrases such as in lines 105 or 356.

Comment 7:  The manuscript was edited and formatted

8- The authors state that before hospital admission, a reliable and straightforward biomarker would have a direct and immediate impact. But this impact is not described further, and practical and theoretical implications are almost missing.

Comment 8: Thanks for this valuable comment, the implications for practice and future research were included in the conclusion section.

9- It seems that a full ethics proposal is missing for this study, although it is a clinical study.

Comment 9: The ethics sentences are put at the end of the manuscript. All ethics documents are sent to the editor.

10- Why are the datasets used in this research's analysis not accessible to the public? This brings a barrier to the reproducibility of the findings by independent research organizations.

Comment 10:  the datasets used in this research's analysis are attached and now accessible to the public. Usually, the datasets are available upon accepted request, but we attach the sheet upon the valuable reviewer’s comment.

11- Are the findings transferable to non-Egyptian patients?

Comment 11: the findings could be transferable to non-Egyptian patients. This study can help clinicians with Egyptian and non-Egyptian patients. EMAP-II is an intracellular polypeptide, expressed upon stress and hypoxia. Also, COVID-19 clinical data and prognosis are almost similar all over the world. So, the findings are transferable to non-Egyptian patients. 

12- The conclusions are very weak. Also, the mortality of COVID-19 patients might be a mortality risk of COVID-19 patients.

Comment 12: The conclusions are written and modulated. Also, the mortality of COVID-19 patients was corrected to be a mortality risk of COVID-19 patients.

Round 2

Reviewer 1 Report

paper is ready

Reviewer 2 Report

The authors have satisfactorily responded to my comments.

Reviewer 3 Report

I thank the authors for improving their manuscript.

Please note that the tables and figures should be formatted consistently. Also, there is a formatting issue with the last reference.